# Detection of toxoplasmic encephalitis in HIV positive patients in urine with hydrogel nanoparticles

Hannah E. Steinberg[1]*, Natalie M. Bowman[2], Andrea Diestra[3], Cusi Ferradas[3], Paul Russo[4], Daniel E. Clark[5], Deanna Zhu[6], Ruben Magni[4], Edith Malaga[3], Monica Diaz[7], Viviana Pinedo-Cancino[8], Cesar Ramal Asayag[9,10], Maritza Calderón[3], Vern B. Carruthers[11], Lance A. Liotta[4], Robert H. Gilman[12], Alessandra Luchini[4], the Toxoplasmosis working group in Peru and Bolivia[¶]

1 Department of Microbiology and Immunology, University of Illinois Chicago, Chicago, Illinois, United States of America, 2 Division of Infectious Disease, School of Medicine, University of North Carolina, Chapel Hill, North Carolina, United States of America, 3 Laboratorio de Investigación en Enfermedades Infecciosas, Universidad Peruana Cayetano Heredia, Lima, Peru, 4 Center for Applied Proteomics and Molecular Medicine, George Mason University, Virginia, United States of America, 5 Vanderbilt University Medical Center, Division of Cardiovascular Medicine, Nashville, Tennessee, United States of America, 6 Department of Epidemiology, University of North Carolina, Chapel Hill, North Carolina, United States of America, 7 Department of Neurology, University of North Carolina, Chapel Hill, North Carolina, United States of America, 8 Laboratorio de Investigación de Productos Naturales Antiparasitarios de la Amazonía, Facultad de Medicina Humana, Universidad Nacional de la Amazonía Peruana, Iquitos, Peru, 9 Universidad Nacional de la Amazonía Peruana, Iquitos, Peru, 10 Department of Infectious Diseases, Hospital Regional de Loreto, Iquitos, Peru, 11 Department of Microbiology and Immunology, University of Michigan Medical School, Ann Arbor, Michigan, United States of America, 12 Department of International Health, Bloomberg School of Public Health, Johns Hopkins University, Baltimore, Maryland, United States of America

¶ Membership of the Toxoplasmosis Working Group in Peru and Bolivia is listed in the Acknowledgments
* hstein5@uic.edu

**Data Availability Statement:** All relevant data are within the manuscript and its Supporting information files.

## Abstract

### Background

Diagnosis of toxoplasmic encephalitis (TE) is challenging under the best clinical circumstances. The poor clinical sensitivity of quantitative polymerase chain reaction (qPCR) for *Toxoplasma* in blood and CSF and the limited availability of molecular diagnostics and imaging technology leaves clinicians in resource-limited settings with few options other than empiric treatment.

### Methology/principle findings

Here we describe proof of concept for a novel urine diagnostics for TE using Poly-N-Isopropylacrylamide nanoparticles dyed with Reactive Blue-221 to concentrate antigens, substantially increasing the limit of detection. After nanoparticle-concentration, a standard western blotting technique with a monoclonal antibody was used for antigen detection. Limit of detection was 7.8pg/ml and 31.3pg/ml of *T. gondii* antigens GRA1 and SAG1, respectively. To characterize this diagnostic approach, 164 hospitalized HIV-infected patients with neurological symptoms compatible with TE were tested for 1) *T. gondii* serology (121/147, positive samples/total samples tested), 2) qPCR in cerebrospinal fluid (11/41), 3) qPCR in blood (10/

**Funding:** Robert H. Gilman supported staff salaries and laboratory supplies in this investigation with NIAID R01 (AI136722-01). Robert H. Gilman also supported the training of many of the Peruvian and Bolivian authors and working group members with 1D43TW010074-01. Natalie M. Bowman supported sample collection and was supported by a NIAID K23 award (AI113197), UNC CFAR P30 AI50410, Fogarty R25TW009340, and Burroughs Wellcome Fund/ASTMH Postdoctoral Fellowship in Tropical Infectious Diseases. Alessandra Luchini supported staff salaries and laboratory supplies in this investigation with NIAID R21 (AI138135). Lance A. Loitta supported staff salaries and laboratory supplies in this investigation with NICHD R21 (HD0974720). Hannah E. Steinberg was supported by NIH Fogarty International Center Grant #D43TW009340. Cusi Ferradas and Monica Diaz were supported by the Fogarty International Center of the National Institutes of Health under Award Number D43TW009343 and the University of California Global Health Institute. The funders did not play any role in the study design, data collection and analysis, the decision to publish or the preparation of manuscript.

**Competing interests:** I have read the journal's policy and the authors of this manuscript have the following competing interests: Alessandra Luchini and Lance A. Liotta are inventors on patents US9,012,240 and US8,497,137 related to the affinity particles. Ceres Nanosciences licensed the rights of these patents that are owned by George Mason University. Alessandra Luchini and Lance A. Liotta own shares of Ceres Nanosciences.

112), and 4) urinary GRA1 (30/164) and SAG1 (12/164). GRA1 appears to be superior to SAG1 for detection of TE antigens in urine. Fifty-one HIV-infected, *T. gondii* seropositive but asymptomatic persons all tested negative by nanoparticle western blot and blood qPCR, suggesting the test has good specificity for TE for both GRA1 and SAG1. In a subgroup of 44 patients, urine samples were assayed with mass spectrometry parallel-reaction-monitoring (PRM) for the presence of *T. gondii* antigens. PRM identified antigens in 8 samples, 6 of which were concordant with the urine diagnostic.

## Conclusion/significances

Our results demonstrate nanoparticle technology's potential for a noninvasive diagnostic test for TE. Moving forward, GRA1 is a promising target for antigen based diagnostics for TE.

### Author summary

Toxoplasmic Encephalitis is a debilitating, yet highly treatable illness, classically seen in person living with HIV lacking treatment. Prompt diagnosis ensures the best outcome possible for patients, but remains a challenge: requiring invasive specimen collection, lacking necessary clinical sensitivity, demanding significant technical skills, and substantial infrastructure. Here we offer proof of concept of a diagnostic approach that is minimally invasive, using a urine-based approach that concentrates *T. gondii* antigens with hydrogel mesh nanoparticles to improve analytical sensitivity for detection by western blot.

## Introduction

Toxoplasma encephalitis (TE) is the most commonly reported neurological opportunistic infection in HIV-infected patients since the introduction of combination ART (cART) [1,2]. Immunocompromised patients infected with *T. gondii* may present with fever, headache, lethargy, incoordination, ataxia, hemiparesis, memory loss, dementia, or seizures [1]. Quantitative Real Time Polymerase chain reaction (qPCR) of cerebral spinal fluid (CSF), considered a reference standard diagnostic test, has a clinical sensitivity of 12%-70% and a specificity of nearly 100% [3–5]. qPCR of blood has poor clinical sensitivity of only 1.5%-35.5% in patients with TE [6–8]. Stereotactic brain biopsy with subsequent organism visualization on pathology provides a definitive diagnosis, but is infrequently done because it is highly invasive and requires neurosurgical services [9,10]. Diagnosis by parasite culture requires 6 weeks, rendering it impractical for clinical use. In many cases, clinical judgment and diagnosis by exclusion are the only options. For patients who are immunocompromised, *T. gondii* infection is life threatening [1], but when treated early, TE has a 90% clinical response rate [11].

Antigens from pathogens can be found in urine, blood, or CSF at very low concentrations, but they are typically masked by abundant native proteins and subject to rapid degradation. Hydrogel nanoparticles increase diagnostic analytical sensitivity by concentrating antigens using semi-specific chemical dye baits with high affinity to target antigens [12–15]. Our group developed a nanoparticle-concentrated urinary antigen detection test for *T. gondii* based on a hydrogel core that captures parasite antigens and excludes interfering high molecular weight proteins [14,15]. Previously, we demonstrated this approach's ability to detect specific antigens

in a tachyzoite lysate matrix and the urine of *T. gondii* infected mice [16]. However, this technique has not previously been tested on human specimens.

The best antigen for detection of acute *T. gondii* infection has not been defined. *T. gondii* has a complex life cycle. Tachyzoites, characterized by rapid proliferation in host cells, are responsible for primary acute disease and pathology from reactivation of latent infection [17]. SAG1 is often used as a diagnostic antigen because of its abundant presence as a glycosylphosphatidylinositol (GPI)-anchored surface protein on tachyzoites [18–20]. Less explored is dense granule protein 1(GRA1), which is secreted in high quantities shortly after the parasite enters a host cell [21–23].

In this report, we describe assays utilizing nanoparticles to capture, concentrate, and detect *T. gondii* antigens SAG1 and GRA1 from urine. Urine from 215 HIV-infected persons (164 hospitalized, 51 ambulatory) in Peru and Bolivia were tested for the presence GRA1 and SAG1 using the nanoparticle capture technique in combination with western blot immunoassay. The blood of these patients was tested for *Toxoplasma*-specific IgG and circulating *T. gondii* parasites by qPCR using primers for repeat element 529 [24]. Because clinical sensitivity of qPCR in blood for *T. gondii* is notoriously poor, we used qPCR in CSF when available. Two additional techniques were employed to further validate the urine nanoparticle technique: parallel-reaction-monitoring (PRM) mass spectrometry for GRA1 and SAG1 in urine and qPCR in CSF. This proof of concept work demonstrates the potential of hydrogel nanoparticle-based technology to produce more sensitive and specific diagnostic tests for TE in HIV-infected patients.

## Methods

### Ethics statement

The institutional review boards of study hospitals and involved institutions approved patient collection protocols. In Bolivia: Universidad Católica Boliviana "San Pablo", Santa Cruz; Hospital Clínico Viedma, Cochabamba; Instituto de Desarrollo Humano, Cochabamba; Colectivo de Estudios Aplicados, Desarrollo Social, Salud y Medio Ambiente, Cochabamba. In Peru: Hospital Regional de Loreto, Iquitos; Hospital Dos de Mayo, Lima; Asociación Benéfica Prisma, Lima; Universidad Peruana Cayetano Heredia, Lima. In the United States: University of North Carolina, Chapel Hill, NC, and Johns Hopkins University, Baltimore, MD. The IRB at George Mason University, VA approved the collection of control urine from healthy participants. Formal written consent was obtained for all participants, in cases in which the participant themself was unable to provide consent, their health care proxy did so.

**Study population.** Hospitalized HIV positive participants: Inclusion criteria included confirmed HIV infection, age at least 18 years, and informed consent. One hundred and sixty four HIV positive patients from Santa Cruz (SC) and Cochabamba (CBBA), Bolivia and Iquitos (IQ) and Lima (LIM), Peru were recruited. Patients recruited in the Peruvian sites presented with neurological symptoms. Patients recruited in Bolivia were part of broader studies of persons living with HIV (PLHIV); they were included in this study if they presented with neurological symptoms and they (or their legal guardian) provided written informed consent.

Asymptomatic HIV/*T. gondii* seropositive participants: 51 ambulatory HIV/*T. gondii* positive patients for controls were recruited from PROCETSS (Programa de Control de Enfermedades de Transmisión Sexual y Síndrome de Inmunodeficiencia Adquirida) at Hospital Regional de Loreto in Iquitos, Peru. Participant inclusion criteria included confirmed HIV infection, age at least 18 years, lack of current neurological symptoms, and informed written consent.

**Clinical procedures.** Blood and urine specimens were taken at enrollment, which for most hospitalized patients was shortly after admission; remnant CSF was collected if the subject underwent lumbar puncture as part of their medical care. Urine and CSF were refrigerated immediately and frozen (at -20˚C or -80˚C depending on the collection location) within 24 hours. Blood was collected without additive, coagulated for 1 hour, and centrifuged to separate serum and clot, which were separated then frozen (at -20˚C or -80˚C depending on the collection location) until analyzed. CD4 and CD8 cell counts and viral loads were abstracted from participants' charts. A questionnaire of demographic and clinical data was administered to each hospitalized subject or their health care proxy if their neurologic status was impaired.

**Nanoparticle production.** Reactions were carried out in 1000mL round-bottom flask using a condenser. Nine grams (g) N-Isopropylacrylamine (NIPA- Sigma) and 0.280 g N,N' methylenbisacrylamide (BIS- Sigma M7279) were dissolved in 250 mL of water and filtered-nitrocellulose membrane (0.45μm, Millipore). The system was purged with nitrogen for 30 minutes. 667 μL of allylamine (Sigma) were added to the solution. The solution was held under nitrogen for 15 minutes then heated to 70–80˚C for 30 minutes. Then 0.1g potassium persulfate (Sigma) was dissolved in 5 mL of water and injected into the solution. The reaction was stirred at 70–80˚C for 6 hours. The nanoparticles were then washed 4–5 times at 19000g for 40–50 minutes [12,14,15].

**Nanoparticle dying.** Reactive blue-221(RB221) dye was prepared by dissolving 3.96g of $NaHCO_3$ in 240mL of MilliQ water. 1.8 g of RB-221 dye was added to the solution with stirring. Dye solution was centrifuged for 30 minutes at 19000g. Supernatant was filtered (0.45μm Millipore). Dye solution and nanoparticles were combined and stirred overnight. Dyed nanoparticles were then washed by resuspension in milliQ water and then pelleted at 19000g for 40–50 minutes. Washing was repeated until the supernatant appeared clear; approximately 4–5 washes were required.

**Nanoparticle incubation with urine.** To reduce variability of the assay, pH of urine samples was adjusted with 1M HCl to a pH between 5 and 6. Urine was centrifuged at 3500g for 10 minutes to remove cellular debris, and supernatant was saved. Positive and negative control samples were produced by: collecting control urine from healthy lab members at George Mason University and adjusting the pH to be between 5 and 6; centrifuging at 3500g for 10 minutes to remove cellular debris and saving the supernatant. For the positive control specified concentration of recombinant protein or tachyzoites were spiked into the control urine. Nanoparticles were added in proportion to the urine supernatant volume (20% of total volume). Urine and nanoparticles were incubated for 30 minutes. The solution was centrifuged at 13000g for 15 minutes. Supernatant was removed and nanoparticles were washed in 1ml water. After centrifugation, the nanoparticle pellets were eluted with Novex sample loading buffer. Elution occurred at 100˚C for 10 minutes and was scaled (0.1X) to original nanoparticle volume. If elution volume was greater than 40μL (the maximum volume a gel well can hold), tubes were left open during heating to allow for concentration of eluate for gel electrophoresis. A total protein loading control of equivalent concentration to the urine spike in control was prepared by adding either recombinant protein or tachyzoites to 10ul of elution buffer.

**SDS PAGE.** After elution, samples were centrifuged at 13000g for 10 minutes. Supernatants were loaded onto a 4–20% Tris-glycine gel (Life Technologies) and run for 1 hour at 200V in 1X running buffer (Life Technologies) in a Novex X-Cell IITM Mini-Cell apparatus (ThermoFisher).

**Western blot.** After 1-D gel separation, gels were transferred (in a wet tank) to a PVDF membrane (BioRad) for 1.5 hours at 35V. Transferred membranes were blocked for 30

minutes in 5% dry-milk PBS-Tween. Membranes were incubated overnight at 4˚C with rocking in monoclonal mouse anti-SAG1 (ThermoFisher, Clone P30/3) 1:500 in 5% dry milk-PBS-Tween, or for 1 hour with mouse monoclonal antibody Tg17-43 anti-GRA1 (provided by Dr. Cesbron-Delauw) [25]. Membranes were washed and incubated with HRP conjugated goat anti-mouse antibody (Invitrogen) 1:10,000 in 5% dry milk-PBS-Tween. Membranes were subjected to Supersignal WestDura (ThermoFisher) for 5 minutes before exposure.

**Serology.** *Toxoplasma gondii* IgG serological status was determined with an in-house enzyme linked immunosorbent assay (ELISA). The protocol for in-house ELISA was assessed against IBL International's *Toxoplasma gondii* IgG enzyme linked immunosorbent kit. For this assessment, a total of 38 serum samples were tested, of which 19 samples from healthy volunteers and 19 serum samples were from PLHIV. The samples were run in duplicate to determine reproducibility and controls were included to ensure the quality of the results. Compared to IBL International's kit, the in-house ELISA had a clinical sensitivity of 100%, 95%CI (86.3%-100%), specificity of 92.3%, 95%CI (64%-99.8%), kappa index of 0.94, 95% CI (0.83–1.06), and a concordance index of 0.97. Of note, IBL International's kit has not been certified as a diagnostic test, but is produced by a CE accredited manufacturer.

For the in-house ELISA, Nunc MaxiSorp (Thermo Fisher) plates were coated with 100μL (1ug/ml) of Total Lysate Antigen (TLA) overnight at 4˚C. The plate was washed five times with PBS-T, blocked with 5% dry-milk PBS-T for two hours, followed by five washes of PBS-T. Patient serum was diluted 1:500 in 1% dry-milk PBS-T and incubated for one hour at 37˚C. The plate was washed five times with PBS-T. Anti-Human IgG (H+L) HRP (SeraCare) diluted 1:10,000 in 1% dry-milk PBS-T was incubated for one hour at 37˚C. The plate was washed three times with PBS-T and two times with PBS 1x. TMB Microwell Peroxidase Substrate System (SeraCare) was added for two minutes prior to stopping with 2M sulfuric acid. The plate was read at 450nm wavelength.

**qPCR.** Blood clot specimens were homogenized with 300ul guanidine hydrochloride 6M (Sigma-Aldrich, USA). After homogenization, the sample was transferred to a Lysing Matrix H 2 mL tube (MP Biomedicals, USA) followed by an agitation cycle in a FastPrep-24 5G machine (MP Biomedicals, USA) (5.5 m/s– 30") to ensure the clot disaggregation before the treatment with proteinase K [26]. DNA was then extracted from treated clot specimen and CSF specimens using the High Pure PCR Template Preparation Kit (Roche Life Science) per manufacturer's instructions. Target sequences were amplified using a Light Cycler (Applied Biosciences) using the following program: 2 minutes at 50˚C, then 10 minutes at 95˚C; and 40 cycles of 95˚C for 15 seconds, 60 seconds at 58˚C, and 60 seconds at 72˚C; followed by a 4˚C hold. Primers for 529 repeat were 5′- GCTCCTCCAGCCGTCTTG (forward) and 5′- TCCTCACCCTCGCCTTCAT (reverse), and FAM—AGGAGAGATATCAGGACTGTA— 3'MGB probe was used [24,26]. Positive controls consist of a pool of positive patient samples and a standard logarithmic curve of RH strain tachyzoites from 1–100000 [26].

**Culture of T. gondii isolates strain.** Isolates were maintained by successive passage in LLC-MK$_2$ (*Macaca mulatta*, ATCC/CCL-7) in media: RPMI 1640 (Sigma), 2% heat-inactivated fetal bovine serum (Gibco), pyruvate (1mM; Gibco), non-essential amino acid mixture 1:100 (Gibco), gentamicin (45μg/mL; Gibco), and penicillin (100U/mL; Gibco), incubated at 37˚C and 5%CO$_2$. Growth media was changed every four days and cultures were passaged every ten days.

**Preparation of T. gondii lysates.** The tachyzoites from culture of the RH strain were washed three times in PBS and sonicated: 4 cycles, 4Hz, 30 seconds, with one-minute rests. Lysates were centrifuged at 13,000g for 15 minutes at 4˚C.

**Mass spectrometry.** Prior to PRM analysis, seven different *T. gondii* isolates from BEI resources were trypsin-digested, then analyzed by LC-MS/MS, identifying 1,054 non-redundant proteins (Supplemental Table 1 of *Steinberg et al* 2016 [16]). Thirty-four peptides, representing

all 12 proteins, were targets for PRM because of their protein ion fragmentation patterns. Forty-four patients were selected for MS analysis based on their western blot and qPCR results: We selected a mix samples from participants that were positive across multiple measures, negative across multiple measures, and discordant between different measures. Positive controls were lysates from whole cultured tachyzoites, recombinant SAG1 and GRA1 that were spiked into urine. 2.5–3 ml of urine from these patients (S1 Table) were incubated 30 minutes with the nanoparticles. Nanoparticles were washed once with water, and antigen was eluted at 100˚C with 1% Rapigest in 50mM ammonium bicarbonate and 10mM TCEP. Eluate was alkylated with 50mM iodoacetamide, then digested with trypsin overnight at 37˚C. Digestion was halted using trifluoroacetic acid (final concentration 0.1%). Samples were desalted with C-18 spin columns (ThermoFisher), dried by vacuum centrifugation, and reconstituted in 0.1% formic acid.

Digested samples were analyzed by PRM on an Orbitrap Fusion mass spectrometer (ThermoFisher) with a nanospray EASY-nLC 1200 HPLC (ThermoFisher). Peptides were separated using a reversed-phase PepMap RSLC 75μm i.d. × 15cm long with 2μm, C18 resin LC column (ThermoFisher). The mobile phase was 0.1% aqueous formic acid (mobile phase A) and 0.1% formic acid in 80% acetonitrile (mobile phase B). After sample injection, the peptides were eluted using a linear gradient (5%-50%B) over 15 min and ramping to 100%B for an additional 2 min. The flow rate was set at 300nL/min. Data were analyzed with Skyline v3.6 to determine the presence or absence of proteins/peptides of interest.

**Statistical analysis.** Data were analyzed using STATA13 (StataCorp LLC, College Station, TX). Baseline characteristics were compared by student T-test, chi-squared analysis, and logistic regression as appropriate. Comparison of repeated qPCR runs was performed using a paired McNemar's test or a paired Cochran's q-test depending on the number of repeated analyses.

## Results

### Characterization of the HIV-infected patient population

Fig 1 illustrates the flow of study activities. Of the 228 participants that met inclusion criteria, thirteen were excluded from the study because of absent urine sample. The remaining 215 participants fell into two groups, ambulatory (n = 51) and hospitalized (n = 164). Table 1 describes the study participants. Of the hospitalized patients, 19 patients were from Hospital Clínico Viedma, Cochabamba; 32 were from San Juan de Dios, Santa Cruz; 66 were from Hospital Regional de Loreto, Iquitos; and five were from Hospital Dos de Mayo, Lima. Hospitalized and ambulatory patients did not differ by age or sex. Hospitalized participants had significantly lower CD4 counts (median 90, p<0.001) and higher viral loads (median 50,380, p = 0.005) than the ambulatory group (median CD4 430, median viral load 0). The seroprevalence of *T. gondii* in the hospitalized group was 82% and 100% in the ambulatory group.

### Quantitative PCR

One hundred and seventy three patients' blood samples and 41 patients' CSF samples were available for *T. gondii* qPCR. *T. gondii* DNA was detected by qPCR in 8% (10/122) of the hospitalized participants' blood samples and 27% (11/41) of their CSF samples.

### Nanoparticles significantly improved analytical sensitivity of recombinant antigen detection

Prior to employing the nanoparticle western blot to detect *T. gondii* antigen in patient urine samples, we assessed the nanoparticle western blot's dynamic range. Fig 2 illustrates that recombinant glutathione S-transferase (GST) tagged-GRA1 and polyhistidine (HIS) tagged-

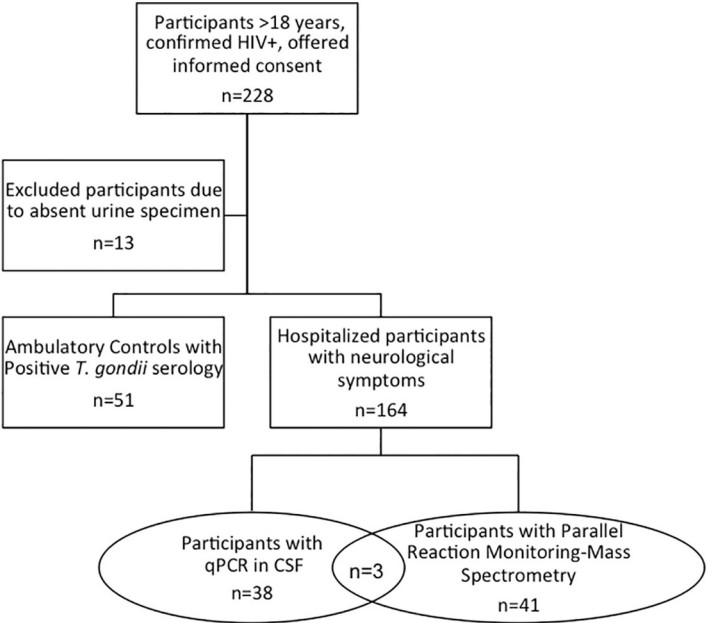

**Fig 1. Flow chart of study groups and tests performed.**

SAG1 antigens are undetectable at concentrations less than 500pg/ml when spiked into urine without nanoparticle incubation; with nanoparticles, antigens can be detected at concentrations of 31.3pg/ml for SAG1 and 7.8pg/ml for GRA1 (Fig 2). S1 Fig demonstrates the limit of detection of the western blots for GRA1 (1.95ng/ml) and SAG1 (125ng/ml) without nanoparticle concentration.

## Use of nanoparticle western for antigen detection in the urine of HIV positive persons

Fig 3A and 3B shows representative results from urine from HIV positive hospitalized patients tested for SAG1 and GRA1, respectively. Fig 3A demonstrates that after nanoparticle

**Table 1. Description of the study population: Participant characteristics as summarized by basic demographic, geographic, and laboratory studies.**

|  | Hospitalized HIV+ Participants [n = 164] | Ambulatory HIV+ Participants [n = 51] | Odds Ratio (95%CI) |
|---|---|---|---|
| Median Age (IQR) | 34 (28–43) | 38 (30–47) | 0.98 (0.95–1.01) |
| Sex n = 202 |  |  |  |
| Male (%) | 112 (73%) | 32 (65%) | 0.71 (0.35–1.40) |
| Female (%) | 42 (27%) | 17 (35%) |  |
| CD4 Count-Median (IQR) | 90 (47–190) | 430 (306–637) |  |
| HIV Viral Load- Median (IQR) | 45,061 (5,608–284,669) | 0 (0–245) |  |
| Positive *T. gondii* Serology n = 176 | 122 (82%) | 51 (100%) |  |
| Self Reported Antiretroviral Usage n = 160 | 61 (56%) | 51 (100%) |  |
| Cochabamba, Bolivia | 51 (31%) | - |  |
| Santa Cruz, Bolivia | 40 (24%) | - |  |
| Iquitos, Peru | 68 (41%) | 51 (100%) |  |
| Lima, Peru | 5 (3%) | - |  |

Sample sizes- Median Age: hospitalized n = 151, ambulatory n = 49; CD4 Count: hospitalized n = 114, ambulatory n = 51; HIV Viral Load: hospitalized n = 87, ambulatory n = 35

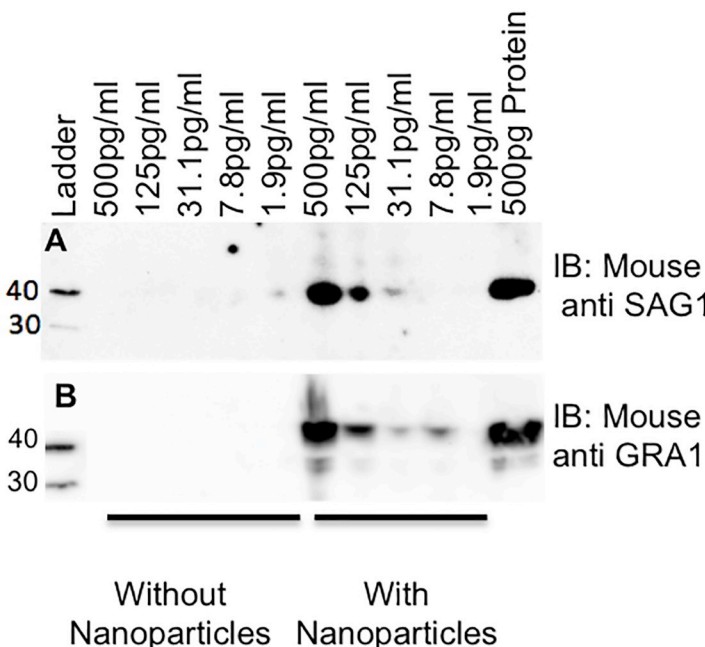

**Fig 2. Nanoparticles significantly increase limits of antigen detection over standard western blot.** (A) Recombinant His-tagged SAG1 and, (B) recombinant GST-Tagged GRA1 were spiked into 20mL of human urine and incubated with and without nanoparticles.

concentration, urine from two patients (CBBA54 and 55) were positive for SAG1 (30kDa) by western blot. Fig 3B shows that urine from three patients (SC139, 140,141) tested positive for GRA1. Two bands are visible, the predicted 26kDa band and one at 32-33kDa, corresponding to membrane bound and secretory GRA1, respectively [27]. The 26kDa band is faint without nanoparticle-concentration in samples SC139 and 141 but is better visualized with the nano-particle concentration step. A faint 33kDa band can only be detected in sample 139 without nanoparticle concentration while its signal is greatly enhanced and visible in all samples after nanoparticle processing. Fifty-one Peruvian asymptomatic *T. gondii*-seropositive, HIV-infected individuals were also tested for GRA1 and SAG1 by urine nanoparticle western blot and blood qPCR. All 51 tested negative.

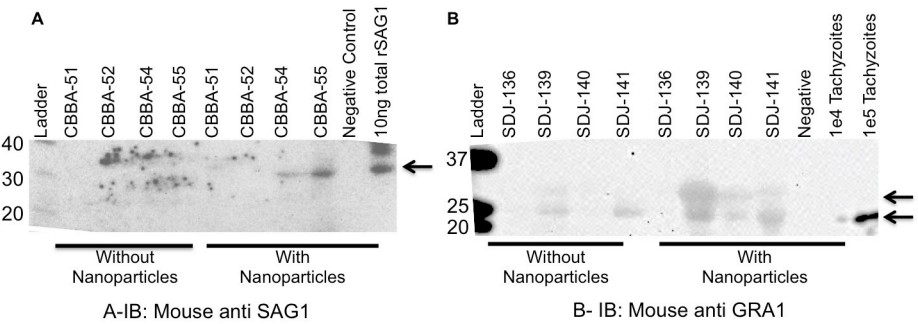

**Fig 3. Test of human urine samples.** One mL of urine was incubated with nanoparticles. Following nanoparticle incubation, nanoparticles were eluted, separated by SDS-PAGE, and transferred to a PVDF membrane. The membrane was then immunoblotted for mouse anti SAG1 (3A) and mouse anti GRA1 (3B). 3A and 3B are representative of the hospitalized group.

**Table 2. qPCR in CSF compared to nanoparticle western blot from CSF in the hospitalized patients.** 2X2 table comparing qPCR in patient CSF to nanoparticle western blot in patient urine.

| | | CSF qPCR | |
|---|---|---|---|
| GRA1 Western Blot Result | | Positive (N = 11) | Negative (N = 30) |
| | Positive | 5 (45%) | 1 (3%) |
| | Negative | 6 (55%) | 29 (97%) |
| SAG1 Western Blot Result | | | |
| | Positive | 0 | 3 (10%) |
| | Negative | 11 (100%) | 27 (90%) |

## Comparison of qPCR and nanoparticle western blot

Table 2 compares the outcomes of the urine nanoparticle western blot for GRA1 and SAG1 to qPCR of CSF. We saw a significant association (p = 0.007) between qPCR of CSF and GRA1 antigen detection in the urine. GRA1 was detected in urine from 5/11 subjects with a positive qPCR of CSF. SAG1 did not perform as well: it could not be detected in any qPCR-positive CSF subjects but was detected in 3 samples from negative subjects. S1 Table compares outcomes of urine nanoparticle western blot for GRA1 and SAG1 to qPCR of blood. We did not see any association between qPCR of blood and neither GRA1 nor SAG1. The differences between the qPCR of blood and CSF have been previously documented in the literature, with qPCR of CSF acknowledged as the superior reference standard [3–8].

## Mass spectrometry analysis of urine and CSF

Thirty-four peptides from 12 proteins were monitored for PRM Mass spectrometry: SAG1, SAG2A, GRA1, GRA3, GRA7, GRA12, ROP4, ROP5, ROP7, RON8, MIC10, and M2AP. Results were considered positive when precursor ion retention times correlated between samples and positive controls, fragmentation peaks, co-eluted and matched in relative intensity the positive controls. Only GRA was detected. Fig 4 displays the representative positive results for GRA1, peptide VERPTGNPDLLK. Fig 4A and 4B are 1 million tachyzoites and 200ng recombinant GRA1 respectively. Fig 4C and 4D are the mass spectra for peptide VERPTGNPDLLK, found in the urine of the listed patients. Fig 4E is the key of the b+ and y+ ions. Seven of eight subjects with identifiable GRA1 peaks tested positive by either western blot antigen (n = 4), blood qPCR (n = 1); both western blot and blood qPCR (n = 1); or western blot, CSF and blood qPCR (n = 1). S2 Table displays the results from all study participants for all assays.

There were three subjects with PRM urine analysis that also had qPCR results available from their CSF. We attempted PRM analysis on urine specimens from additional patients, but unfortunately equipment failure destroyed those specimens. Of the 3 participants with available CSF specimens, 2 participants did not have identifiable peptides in their urine and were negative by qPCR in CSF and the urine nanoparticle western blot, but were *T. gondii* serologically positive. One patient was positive by all 3 tests, PRM, qPCR of CSF, and urine nanoparticle western blot.

## Mass spectrometry analysis of CSF

CSF was available for PRM from two subjects who were both *T. gondii* seropositive and presented with neurological symptoms. The CSF was processed identically to urine samples for PRM analysis including the production of controls. One of the 2 subjects (IQ23) had fragmentation masses and a retention time that matched GRA1. This patient had identical peptides detected in the urine and the CSF and was positive by CSF qPCR and nanoparticle western

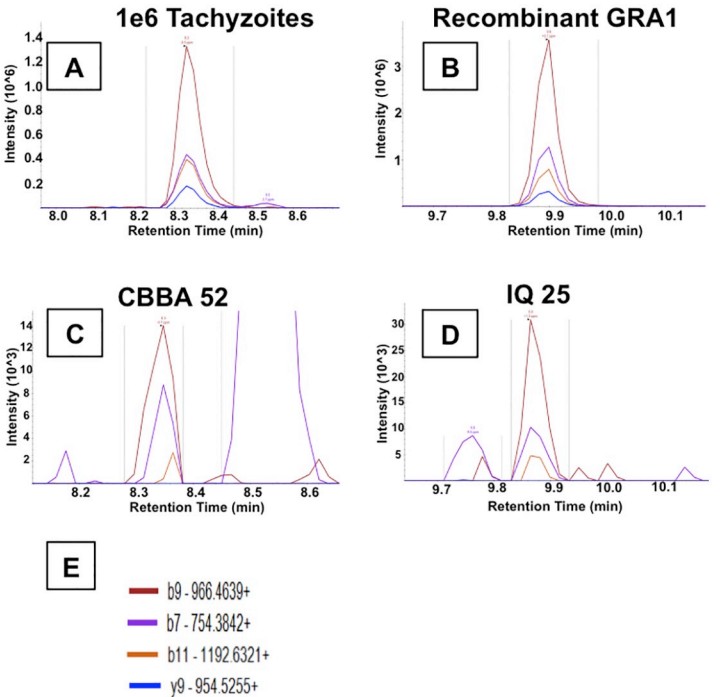

**Fig 4. Positive patient urine by mass spectrometry.** PRM mass spectra showing fragmentation ions (b9, b7, b11, y9) for peptide VERPTGNPDLLK, originating from protein GRA1. A subgroup of 44 urine samples were tested for the presence of 12 *T. gondii* proteins SAG1, SAG2A, GRA1, GRA3, GRA7, GRA12, ROP4, ROP5, ROP7, RON8, MIC10, and M2AP. Only GRA1, peptide VERPTGNPDLLK was found. VERPTGNPDLLK was identified based on the consistent identification of ion b9+, b11+, b7+, and y9+ consistent retention time across controls and samples. b+ ions extend from the amino end of the peptide, while y+ ions extend from the carboxyl end. The associated number is indicative of how many amino acids were associated with each ion. (A) 1e6 RH strain tachyzoites after incubation with nanoparticles acting as a positive control. (B) Recombinant GRA1 protein after incubation with nanoparticles acting as a 2nd positive control. (C&D) Representative urine samples from patients infected with HIV and possible Toxoplasmic Encephalitis CBBA52 and IQ25. All patients that show a positive result for peptide VERPTGNPDLLK, demonstrate the presence of protein GRA1 in their respective urine samples. Representative sample CBBA52 has retention time (~8.3 min) corresponding to the tachyzoite positive control (A), while representative sample IQ-25 has retention time (~9.8 min) corresponding to the recombinant GRA1 positive control (B). The shift in retention times was due to a necessary change in the LC column. (E) The fragmentation ion key for ions b9, b7, b11, y9; each peak in part (C&D) corresponds to an ion of the documented color in E.

blot for GRA1. The other CSF specimen was negative by PRM, qPCR of CSF, and nanoparticle western blot for GRA.

## Discussion

This study offers proof-of-concept that a nanoparticle-based western blot test can detect *T. gondii* antigen in urine from HIV-infected patients with suspected TE. Urine testing could provide a noninvasive alternative to lumbar puncture, which is uncomfortable and can carry a risk of herniation of intracranial contents through the tentorial hiatus or foramen magnum in a patient with a space-occupying brain mass. A urine-based diagnostic may be useful when CSF cannot be obtained or PCR is not available. A urinary antigen test could also be used to screen for reactivated toxoplasmosis in other immunosuppressed populations such as patients with cancer or organ transplants.

CSF PCR is not a gold standard to diagnose TE, thus we could not calculate true clinical sensitivity of the nanoparticle assay. However, CSF PCR is a clinically accepted reference standard because of its specificity. [6–8] The GRA1 assay exhibited an 83% positive percent

agreement and an 83% negative percent agreement as compared to CSF PCR. The lack of positive nanoparticle assays in *T. gondii*-seropositive ambulatory patients with HIV corroborates its specificity and negative percent agreement. Although urine has been used for the diagnosis of toxoplasmosis [28–30], each prior assay was qPCR-based. The majority of these assays have focused on detection of antigen from infected mice [28,31]. Two other groups have attempted to detect active *T. gondii* infection in urine; Fuentes *et al* detected toxoplasmosis in the urine of one congenital case with a PCR based approach [29]. Ayi *et al* applied a membrane-based ELISA approach, which focused on detection of ocular and congenital toxoplasmosis. [32] Our assay uses a novel antigen detection technology and successfully identified at least 5 subjects with qPCR-confirmed TE.

Our study did show that GRA1 performed significantly better than SAG1 to detect *Toxoplasma* antigen in urine. SAG1 has previously been shown to be a useful target for IgG *T. gondii* antibody serology with both high sensitivity and specificity; however, many groups have failed to develop an assay that consistently detects active *T. gondii* infections (such as TE) using SAG1 antigen [20,33–36]. Our assay's poor positive percent agreement for SAG1 may indicate that the antigen is excreted at low abundance or too degraded in the urine for consistent immunoassay detection. SAG1 is highly immunogenic, which may lead to its degradation by phagocytes, in contrast to GRA family proteins, which are associated with TGF-ß production and dampening of the immune response [37,38]. It is also possible that SAG1 is not shed by tachyzoites because it is surface protein, unlike GRA1, which is a secretory protein. A secretory protein may be released in higher amounts from the parasite, while a structural protein could be made only in sufficient quantity for assembly of the parasite surface or remain localized there rather than released into systemic circulation. Alternatively, the different serotypes of *T. gondii* present in South America, where *T. gondii* is particularly diverse, might affect assay performance. More research into the effect of parasite diversity on diagnostic test performance is needed.

PRM is a targeted mass spectrometry analysis that is able to directly detect peptides derived from specific proteins of interest [39,40]. Prior LC-MS/MS experiments identified fragment ions b7, b9, b11, y9 as the best ones for detection of GRA1 peptide VERPTGNPDLLK during PRM analysis based on the abundance of these ions. Previously our group has detected peptide VERPTGNPDLLK of GRA1 in the urine of *T. gondii* infected mice by PRM [16]. Here we used PRM to confirm the presence of antigen from *T. gondii* in patient urine as detected by our nanoparticle-based western blot. We detected GRA1 peptide VERPTGNPDLLK in urine from 8 patients and in the CSF of one. The CSF findings were consistent with both urine nanoparticle western blot and qPCR in CSF. The comparison of patient samples to positive controls demonstrates that: 1) the retention times of VERPTGNPDLLK are the same, 2) the 4 fragment ions have the same relative abundance, and 3) the accuracy of all 4 fragment ions is approximately 1ppm or less. This sub 1ppm accuracy allows discrimination between signals from peptides of interest and those derived from the matrix. Two participants tested negative with the PRM approach but positive with the urine nanoparticle western blot. This discrepancy may be due to different target regions of the anti-GRA1 antibody and PRM. VERPTGNPDLLK may be a region flanking the antibody epitope or a distant sequence.

A limitation of our study was the lack of good historical and clinical data due to severe illness in most inpatients and limited availability of diagnostic testing and past medical records. Data was missing for CD4 counts and viral loads on many patients, usually because of early death before the specimens were collected. Computed Tomography (CT) scans were not obtained on the majority of our subjects, mainly due to financial limitations. Analysis of the participants with heads CTs (n = 57) did not reveal a definitive pattern on imaging. Additionally, very few participants underwent lumbar puncture, which limited our sample size for

comparisons of the nanoparticle assay to the reference standard CSF qPCR. *T. gondii* is an intracellular pathogen, thus detection of its genomic and protein components outside of tissue is likely to be highly inconsistent. All hydrogel nanoparticles were dyed with the same lot of reactive blue 221. This ensured reproducible capture of GRA1 and SAG1. However, there is a high degree of lot-to-lot variability in dye production (especially if dye is sourced from the textile industry). Larger scale and/or commercial use of this test would require standardized dye production. The nanoparticle western blot assay is not technically simple, electrophoresis and western blotting requires a high degree of infrastructure and trained personnel, thus limiting the reach of this approach. Additionally, limited lanes in the electrophoresis step precluded the possibility of quantification of the bands using an internal standard curve. Only 1mL of urine was tested for each patient per antigen to assure consistency across all patients, but the assay can accommodate larger volumes that will likely increase sensitivity. In Chagas and Lyme disease urine nanoparticle diagnostics, larger volumes of urine increased the analytical sensitivity and specificity at least 10-fold. [41–43] Finally, the lack of a true gold standard limits our ability to calculate clinical sensitivity and specificity. CSF qPCR is <70% sensitive, so a definitive diagnosis of TE was impossible in most study participants. By comparing the nanoparticle western to CSF qPCR we maybe overestimating the success of the assay.

Despite these limitations, we are encouraged by this new diagnostic possibility because of the unprecedented detection limits of this assay for *T. gondii* antigens in urine and the confirmation of GRA1's presence in urine and CSF by PRM mass spectrometry analysis. TE diagnostics remain challenging. No current method is sufficiently sensitive and specific to provide clinicians and their patients a reliable diagnosis of TE, often leading them to resort to brain biopsy or empiric treatment. Hydrogel nanoparticle-based antigen detection of GRA1 shows promise for clinical test development for TE. Nanoparticle-based assays could be a valuable screening tool for early detection and treatment of TE that does not require invasive testing or access to expensive molecular diagnostics. Future work will focus on identifying additional antigens for urine testing, improving assay clinical sensitivity, adding a quantitative aspect to the assay, and evaluating the assay in a blinded study in well-defined clinical population. Based on our findings, we are optimistic that with the identification and addition of further antigens to a testing panel, we may be able to successfully diagnose TE safely and non-invasively using urine.

## Supporting information

**S1 Table. Comparison of qPCR of blood with urine nanoparticle western blot.** 2X2 table comparing qPCR in patient blood to nanoparticle western blot in patient urine.
(XLSX)

**S2 Table. Description of all 215 study participants: Age, sex, CD4 count, viral load, *T. gondii* serological status, *T. gondii* qPCR in CSF, urine nanoparticle western blot and mass spectrometry.** (+) Positive Results, (-) Negative Results, (.) Missing Result.
(XLSX)

**S1 Fig. Limit of detection of western blot without nanoparticles: Recombinant GRA1 or recombinant his-tagged SAG1 were spiked into 1mL of urine.**
(TIFF)

## Acknowledgments

We would like to acknowledge Dr. Rafael Saavedra for his kind donations of the RH strain tachyzoites. Mouse monoclonal antibody Tg17-43 anti GRA1 was kindly provided by Dr.

Marie-France Cesbron-Delauw. Drs. Javier Bustos and Hector Garcia kindly provided their assistance reviewing participants for alternate neurological conditions. Dr. Calle for reading the head computed tomography scans. We would like to acknowledge Dra. Lastenia Ruiz Mesía for her contribution and support of this project. We would also like to acknowledge the support for the statistical analysis from the National Center for Research Resources and the National Center for Advancing Translational Sciences (NCATS) of the National Institutes of Health.

Members of the **Toxoplasmosis Working Group in Peru and Bolivia**: Caryn Bern, Eduardo Ticona, Andres Lescano, Juan Jimenez, Mauricio Dorn, Daniela E. Kirwan, Lilia Cabrera, María Vásquez Chasnamote, Rafael Saavedra Langer, Marilly Donayre Urquizo, Linda Chanamé Pinedo, Jeroen Bok, Gaston Pinedo, Melanie Ayachi, Francesca Schiaffino, Renzo Gutierrez-Loli, Melissa Reimer-McAtee, Meredith Holtz, Taryn Clark, Grace Trompeter, Jeong Choi, Omar Gandarilla, Enzo Fortuny, Anne Palumbo, Gerson Galdos, Roni Colanzi, Raquel Mugruza, Cesar Ticona, Paola Rondan, Aliki Traianou, Jonathan Juliano, Steven Meshnick, Faustino Torrico.

## Author Contributions

**Conceptualization:** Hannah E. Steinberg, Natalie M. Bowman, Andrea Diestra, Cusi Ferradas, Deanna Zhu, Maritza Calderón, Vern B. Carruthers, Lance A. Liotta, Robert H. Gilman, Alessandra Luchini.

**Data curation:** Hannah E. Steinberg, Cusi Ferradas.

**Formal analysis:** Hannah E. Steinberg, Natalie M. Bowman, Paul Russo, Monica Diaz.

**Funding acquisition:** Hannah E. Steinberg, Natalie M. Bowman, Cusi Ferradas, Monica Diaz, Maritza Calderón, Robert H. Gilman, Alessandra Luchini.

**Investigation:** Hannah E. Steinberg, Natalie M. Bowman, Andrea Diestra, Cusi Ferradas, Daniel E. Clark, Deanna Zhu, Edith Malaga, Viviana Pinedo-Cancino, Cesar Ramal Asayag, Maritza Calderón, Vern B. Carruthers, Robert H. Gilman, Alessandra Luchini.

**Methodology:** Hannah E. Steinberg, Natalie M. Bowman, Andrea Diestra, Cusi Ferradas, Paul Russo, Daniel E. Clark, Deanna Zhu, Ruben Magni, Edith Malaga, Viviana Pinedo-Cancino, Maritza Calderón, Vern B. Carruthers, Lance A. Liotta, Alessandra Luchini.

**Project administration:** Hannah E. Steinberg, Natalie M. Bowman, Andrea Diestra, Cusi Ferradas, Daniel E. Clark, Maritza Calderón.

**Resources:** Natalie M. Bowman, Viviana Pinedo-Cancino, Vern B. Carruthers, Lance A. Liotta, Robert H. Gilman, Alessandra Luchini.

**Software:** Hannah E. Steinberg, Paul Russo.

**Supervision:** Hannah E. Steinberg, Maritza Calderón, Alessandra Luchini.

**Validation:** Hannah E. Steinberg, Maritza Calderón.

**Visualization:** Hannah E. Steinberg.

**Writing – original draft:** Hannah E. Steinberg.

**Writing – review & editing:** Hannah E. Steinberg, Natalie M. Bowman, Andrea Diestra, Cusi Ferradas, Paul Russo, Daniel E. Clark, Deanna Zhu, Ruben Magni, Edith Malaga, Monica Diaz, Viviana Pinedo-Cancino, Cesar Ramal Asayag, Maritza Calderón, Vern B. Carruthers, Lance A. Liotta, Robert H. Gilman, Alessandra Luchini.

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
