## [Decision Letter · Decision Letter 0]

4 Sep 2020

Dear Ms. Steinberg,

Thank you very much for submitting your manuscript "Detection of Toxoplasmic Encephalitis in HIV Positive Patients in Urine with Hydrogel Nanoparticles." for consideration at PLOS Neglected Tropical Diseases. As with all papers reviewed by the journal, your manuscript was reviewed by members of the editorial board and by several independent reviewers. In light of the reviews (below this email), we would like to invite the resubmission of a significantly-revised version that takes into account the reviewers' comments. 

Specifically it would be important to address the concern that Toxoplasma encephalitis was not definitively diagnosed, bringing into question the significance of this diagnostic test. This can be addressed by limiting the study to patients with evidence of Toxo encephalitis. Further, this method was not compared to a gold-standard. Lastly, reviewers felt that the limitations of the study were not adequately addressed and had additional minor comments that should be addressed.

We cannot make any decision about publication until we have seen the revised manuscript and your response to the reviewers' comments. Your revised manuscript is also likely to be sent to reviewers for further evaluation.

Sincerely,

Photini Sinnis, M.D.

Deputy Editor

Reviewer's Responses to Questions

**Key Review Criteria Required for Acceptance?**

**Methods**

-Are the objectives of the study clearly articulated with a clear testable hypothesis stated?

-Is the study design appropriate to address the stated objectives?

-Is the population clearly described and appropriate for the hypothesis being tested?

-Is the sample size sufficient to ensure adequate power to address the hypothesis being tested?

-Were correct statistical analysis used to support conclusions?

-Are there concerns about ethical or regulatory requirements being met?

Reviewer #1: Unfortunately, do not see in their manuscript any info re their patients meeting the minimum criteria for the diagnosis of toxoplasmic encephalitis (TE). Their patients were HIV infected with neurological symptoms (n= 164), Toxoplasma Ig positive (n= 122, 82%), had low CD4 counts (47-190). However, without additional info such as: brain imaging studies revealing multiple ring enhancing brain lesions, positive response to anti-Toxoplasma Rx, it is quite difficult to assess their true denominator. A suggestion is that the manuscript be revised and resubmitted concentrating only in those patients who had evidence of TE and disseminated toxoplasmosis by CSF and urine PCR, respectively. 

For their 164 HIV infected patients with neurological symptoms, there is no statement regarding their consent.

Reviewer #2: Objectives and study are clearly stated. 

The study design has limitations, mainly in the definition of patients and definition of patients. However the design remains relevant to test the hypothesis.

Line 147 The population is clearly described, but small and with some biaises (i.e: all controls are from the same hospital). The population is sufficient for a proof of concept but does not allow for a precise assesment of the performances of their test. However, testing controls from other facilities would be an improvement

Line 190-191 The authors do not specify if the electrotransfert was performed with a tank, semi-dry or dry methods. Please specify

Line 194 Please confirm if the conjugate is an HRP conjugate

Line 197-203 The population used to assess performances of the in-house test is too small. The performances of the IBL test used in the case definition are, to my knowledge, unpublised. Please add confidence intervals to sensitivity and specificity.

This is of minor impact for the study as serology is a very secondary aspect of their work but this should be noted.

Reviewer #3: Are the objectives of the study clearly articulated with a clear testable hypothesis stated? ==> YES

 -Is the study design appropriate to address the stated objectives? ==> Yes. But this study is not going beyond a first basic proof of principle approach. A blinded confirmation study should be designed, or at least mentioned that such validation of the concept is needed 

 -Is the population clearly described and appropriate for the hypothesis being tested? ==> as proof of principle, this population is sufficient

 -Is the sample size sufficient to ensure adequate power to address the hypothesis being tested? => as proof of principle, this sample size is sufficient

 -Were correct statistical analysis used to support conclusions? ==> cannot comment

 -Are there concerns about ethical or regulatory requirements being met? ==> yes

**Results**

-Does the analysis presented match the analysis plan?

-Are the results clearly and completely presented?

-Are the figures (Tables, Images) of sufficient quality for clarity?

Reviewer #1: See comments above

Reviewer #2: Overall: clear results

Line 269-270 Please add standard deviations to median values.

Line 304-305 Figure 2 is blurry and needs higher resolution

Line 323-324 Same for figure 3

Reviewer #3: Does the analysis presented match the analysis plan? ==> yes

 -Are the results clearly and completely presented? ==> yes, but there are concerns on the methodology

 -Are the figures (Tables, Images) of sufficient quality for clarity? ==> some Figures need re-evaluation, depending on the revision of the document.

concerns related to the result section:

Line 115, 140, 291: qPCR: make differentiation between analytical and clinical sensitivity. From the percentages mentioned, it is suggestive to assume this is about clinical sensitivity. In line 291: samples are tested twice to increase the sensitivity. Clarify and add values for both analytical and clinical sensitivity, including Limit of Detection and quantification range. It is not re-assuring if a sample is negative the first time, and after retesting it becomes positive. This can be interpreted as poor assay design and lack of control elements in the experiment. If a sample is negative, and all controls are present, what is the rational for re-testing? 

Line 172: rewrite sentence

Line 298: at what concentration becomes the spiked antigen detectable when not captured with nanoparticles. In absence of a detectable signal in WB, it looks like the experiment as shown in Figure 2 lacks a control for that part of the experiment. The description on Line 178 – 186 does not give the procedure for the control experiment in the approach without nanoparticles.

Line 319: Figure 3: I am not sure I understand the value of Figure 3A and 3B, and also it is confusing to mention that this is a “representative set” of Peruvian asymptomatic individuals. What is representative on a WB showing only negative results? The value of this Figure 3a and 3b needs more explanation. 

Line 319: Figure 3: Figure 3c and 3d: Is it possible from Figure 2 to quantify the band intensity in Fig3C and 3D .Just in case this is not possible due to the differences in the experimental protocols compared to Figure 2, what is the value of Figure 2? Furthermore, in Fig3D, the bands for - for example SDJ-139 – are also visible in the experiment without nanoparticles. One would assume that the concentration is > 500 pg/ml. Use Figure 2B as a calibrator, the concentration for that sample in Fig 3D (with nanoparticles) is not expected to be higher than 125 pg/ml. The discrepancies in signal intensities needs clarification. 

Line 321 -323: indication of subsections of Figure 3 needs correction (labelled ad 2A to 2D). 

Line 334: Table 2: Since qPCR is used as gold standard, it is advisable to have a Limit of detection. The description of qPCR negative results earlier in the document (line 291) does not give confidence that they are truly negative. Hence the percentages of True Negatives (TN) (97% and 90%) might be biased and influencing Specificity (Line 388). The False Positives (FP) (3% and 10%) are concerning, and again require clarification on analytical sensitivity of both approaches. 

Line 387: “imperfect gold standard” is difficult to grasp. If a test is not a gold standard, then rather make this a “reference method”. The issue on the gold standard needs a systematic approach, see also Line 437.

**Conclusions**

-Are the conclusions supported by the data presented?

-Are the limitations of analysis clearly described?

-Do the authors discuss how these data can be helpful to advance our understanding of the topic under study?

-Is public health relevance addressed?

Reviewer #1: See comments above

Reviewer #2: The first part of the conclusion is too strong given the limitations of the study and should be tempered.

For instance, line 381 Urine testing could/may provide

Also, the WB test studied is complex and rather long to perform. This should be discussed.

Line 397-398 please add reference to the claim

Outside of the remarks, the conclusion does answer the initial questions and limitations are described and public health relevance is addressed

Reviewer #3: -Are the conclusions supported by the data presented? ==> yes

 -Are the limitations of analysis clearly described? ==> possibly. the value of urine testing is possibly overestimated by lack of proper work on the gold standard. Clarification of the analytical sensitivity of the qPCR might give a better understanding of the limitations of the nanoparticle testing. At this moment, the nanoparticle technology concept was attributed a high value because of the limitation on the analytical sensitivity of the gold standard. 

 -Do the authors discuss how these data can be helpful to advance our understanding of the topic under study? ==> yes. 

 -Is public health relevance addressed? ==> yes

**Editorial and Data Presentation Modifications?**

Reviewer #1: N/A

Reviewer #2: The bibliography used as reference is not the best up-to-date in the field but remains relevant.

Reviewer #3: recommendation is: Minor Revision.

**Summary and General Comments**

Reviewer #1: Study population included 164 HIV infected patients presenting with neurological symptoms and 51 HIV infected asymptomatic patients from Bolivia and Peru. It is suggested that authors provide the breakdown of their participants by site. The biggest drawback to this study is the absence of "gold standard" for the diagnosis of toxoplasmic encephalitis (TE). Their 122 HIV infected patients who were also positive for T. gondii IgG and had neurological symptoms were at high risk of having TE but do not have more acceptable diagnostic criteria for the TE diagnosis (e.g. multiple brain occupying and ring enhancing lesions, response to anti-Toxoplasma specific Tx). Without these criteria added and well established in their 112 patient population suspected to have TE it is quite difficult to assess the significance of their findings.

Reviewer #2: Overall interesting work with a novel approach for a complex diagnostic. The technique has limitations (long turn around time, performances that still need to defined) but could provide an interesting new option for the specialized labs.

Reviewer #3: The authors should give better attention to the methodology. The urine-based nanoparticle technology has obviously an enormous advantage, but is should be properly compared to a well developed reference method or gold standard. This is missing. It is understandable that the clinical evaluation of the reference method is problematic, but that does not mean that the analytical approach should be excluded. Therefore, the comparison should be at the analytical level, not the clinical sensitivity level. As the results are presented today, this is mixed.
---

## [Decision Letter · Decision Letter 1]

22 Jan 2021

Dear Ms. Steinberg,

Thank you very much for submitting your manuscript "Detection of Toxoplasmic Encephalitis in HIV Positive Patients in Urine with Hydrogel Nanoparticles." for consideration at PLOS Neglected Tropical Diseases. As with all papers reviewed by the journal, your manuscript was reviewed by members of the editorial board and by several independent reviewers. The reviewers appreciated the attention to an important topic. Based on the reviews, we are likely to accept this manuscript for publication, providing that you modify the manuscript according to the review recommendations. 

Reviewer 1 requested some minor edits that should be addressed. 

Sincerely,

Photini Sinnis

Deputy Editor

Reviewer's Responses to Questions

**Key Review Criteria Required for Acceptance?**

**Methods**

-Are the objectives of the study clearly articulated with a clear testable hypothesis stated?

-Is the study design appropriate to address the stated objectives?

-Is the population clearly described and appropriate for the hypothesis being tested?

-Is the sample size sufficient to ensure adequate power to address the hypothesis being tested?

-Were correct statistical analysis used to support conclusions?

-Are there concerns about ethical or regulatory requirements being met?

Reviewer #2: All concerns previously raised have been answered.

One very minor comment for the authors, with no impact on the study: while the use of CLSI guide might be relevant for validating in-house test it requires to be compared to a validated method and whil IBL is indeed CE-mark I've never heard of it being used in any Toxoplasma reference center in Europe nor seen its evaluation published, thus its performances are unknown.

-Are the objectives of the study clearly articulated with a clear testable hypothesis stated? Yes

-Is the study design appropriate to address the stated objectives? Yes

-Is the population clearly described and appropriate for the hypothesis being tested? Yes

-Is the sample size sufficient to ensure adequate power to address the hypothesis being tested? For a proof of concept, yes

-Were correct statistical analysis used to support conclusions? Yes

-Are there concerns about ethical or regulatory requirements being met? No

**Results**

-Does the analysis presented match the analysis plan?

-Are the results clearly and completely presented?

-Are the figures (Tables, Images) of sufficient quality for clarity?

Reviewer #2: All issues previously raised have been answered.

-Does the analysis presented match the analysis plan? Yes

-Are the results clearly and completely presented? Yes 

-Are the figures (Tables, Images) of sufficient quality for clarity? Yes

**Conclusions**

-Are the conclusions supported by the data presented?

-Are the limitations of analysis clearly described?

-Do the authors discuss how these data can be helpful to advance our understanding of the topic under study?

-Is public health relevance addressed?

Reviewer #2: One minor comment left. While this comment requires some rewriting of the conclusion, it does not alter the overall findings.

Line 383-385

"CSF PCR is not a gold standard to diagnose TE, thus we could not calculate true clinical sensitivity of the nanoparticle assay. However, CSF PCR is a clinically accepted reference standard because of its specificity. The GRA1 assay exhibited and excellent specificity of 97% compared to CSF PCR."

I disagree. With a 100% specificity on the qPCR, any positive result is a true positive, allowing correct determination of sensitivity (%urine test positive/qPCR positives). However, a negative qPCR can be either a true or a false negative and a positive urine test with negative qPCR might as well be a true or a false positive, hence specificity cannot be calculated.

It would be best to use positive/negative percent agreement, or reinterpret results according to the previous remark (GRA1 has a low sensitivity compared to qPCR and a negative percent agreement of 97%: this could either be due to a sample detected by urine test and not qPCR or a false positive; SAG1 seems of no interest).

Line 393-396 poor sensitivity -> in fact only negatives results when qPCR is positive. Does not change the rest of the section as SAG1 seems indeed of no interest as stated by the authors

All other concerns previously raised have been answered.

-Are the conclusions supported by the data presented? Outside of the comment above, yes

-Are the limitations of analysis clearly described? Yes

-Do the authors discuss how these data can be helpful to advance our understanding of the topic under study? Yes

-Is public health relevance addressed? Yes

**Editorial and Data Presentation Modifications?**

Reviewer #2: N/A

**Summary and General Comments**

Reviewer #2: There's been an improvement of the overall quality of the presentation of the study, with good efforts made to answers various remarks from reviewers.

PLOS authors have the option to publish the peer review history of their article (what does this mean?). If published, this will include your full peer review and any attached files.

Reviewer #2: Yes: Raphaël P. Piarroux
---

## [Editor Report · Decision Letter 2]

2 Feb 2021

Dear Ms. Steinberg,

We are pleased to inform you that your manuscript 'Detection of Toxoplasmic Encephalitis in HIV Positive Patients in Urine with Hydrogel Nanoparticles.' has been provisionally accepted for publication in PLOS Neglected Tropical Diseases.

Best regards,

Photini Sinnis, M.D.

Deputy Editor

---

## [Editor Report · Acceptance letter]

23 Feb 2021

Dear Ms. Steinberg,

We are delighted to inform you that your manuscript, "Detection of Toxoplasmic Encephalitis in HIV Positive Patients in Urine with Hydrogel Nanoparticles," has been formally accepted for publication in PLOS Neglected Tropical Diseases.

Best regards,

Shaden Kamhawi

co-Editor-in-Chief

Paul Brindley

co-Editor-in-Chief
